**Subject Category:**
Biology (whole organism)

behaviour

schooling, elongation, speed, polarization, transfer entropy, information

**Author for correspondence:**
Maud I. A. Kent
e-mail: maud.kent@sydney.edu.au

†Joint first authors.

# Speed-mediated properties of schooling

Maud I. A. Kent[1,†], Ryan Lukeman[2,†], Joseph T. Lizier[3] and Ashley J. W. Ward[1]

[1]School of Life and Environmental Sciences, University of Sydney, Sydney, New South Wales, Australia
[2]Department of Mathematics, Statistics, and Computer Science, St. Francis Xavier University, Antigonish, Nova Scotia, Canada B2G 2W5
[3]Complex Systems Research Group, Faculty of Engineering & IT, Centre for Complex Systems, The University of Sydney, Sydney, Australia

MIAK, 0000-0001-9922-6189; RL, 0000-0001-8210-7685; AJWW, 0000-0003-0842-533X

Collectively moving animals often display a high degree of synchronization and cohesive group-level formations, such as elongated schools of fish. These global patterns emerge as the result of localized rules of interactions. However, the exact relationship between speed, polarization, neighbour positioning and group structure has produced conflicting results and is largely limited to modelling approaches. This hinders our ability to understand how information spreads between individuals, which may determine the collective functioning of groups. We tested how speed interacts with polarization and positional composition to produce the elongation observed in moving groups of fish as well as how this impacts information flow between individuals. At the local level, we found that increases in speed led to increases in alignment and shifts from lateral to linear neighbour positioning. At the global level, these increases in linear neighbour positioning resulted in elongation of the group. Furthermore, mean pairwise transfer entropy increased with speed and alignment, implying an adaptive value to forming faster, more polarized and linear groups. Ultimately, this research provides vital insight into the mechanisms underlying the elongation of moving animal groups and highlights the functional significance of cohesive and coordinated movement.

## 1. Introduction

Living in a group can improve an individual's foraging efficiency, increase access to mates, enhance the likelihood of detecting a predator and provide benefits such as collective defence, dilution and confusion effects [1]. To retain the benefits of living in a group, animals must move as a group. The resulting collective motion, exemplified by a swarm of insects, a

flock of birds or a school of fish, is often characterized by a surprising degree of coordination and synchronization.

Remarkably, these cohesive group-level patterns emerge as the result of localized rules of interactions. Generally, individuals avoid collisions by moving away from group members that are too close and maintain cohesion by moving towards group members that are too far away. At intermediate distances, these repulsion and attraction forces interact to promote orientation and alignment with neighbours [2,3]. Interestingly, individuals adjust distance and alignment with neighbours primarily through changes in speed, which points to the importance of speed in structuring collective movement and group morphology more generally. In fact, the relationship between speed and collective motion in animal groups has formed the basis of recent work (e.g. Pettit *et al.* [4] and Jolles *et al.* [5]).

As it pertains to the group structure, speed tends to affect an individual's alignment with its neighbours. Specifically, fast-moving groups tend to be more polarized [6]. This relationship between speed and polarization in groups is partly explained by the need to avoid collisions between fast-moving individuals, necessitating a shift from low alignment to high alignment as group speed increases [7]. More broadly, speed is key to understanding shifts between different collective states, such as the shift from loosely polarized shoals, or mills, to highly polarized schools or swarms. Tunstrom *et al.* [8] assert that these different collective states can be characterized broadly using global properties, such as polarization and the degree of rotation, but the shifts between these states are mediated by changes in speed.

Theoretical models have made considerable progress in examining the importance of speed to both local rules of interactions and global properties of the group. For instance, Hemelrijk & Hildenbrandt [9] created models of collective behaviour to demonstrate how specific group morphologies, such as oblong schools, can arise through specific interactions between individuals. They asserted that as individuals slow down to avoid colliding with individuals in front of them, individuals who had previously been neighbours move in to fill the gap, resulting in a narrowing and elongation of the school. This simple mechanistic explanation led to two hypotheses, (1) that groups with more members are denser and more elongate due to the greater frequency of collision avoidance and (2) that slower groups would become more elongate due to lower polarization necessitating greater frequency of collision avoidance. These hypotheses have additional empirical and theoretical support [10,11]. For instance, work on saithe by Partridge *et al.* [12] found greater elongation in slower schools and more circular group formations at higher speeds.

However, Breder [13] predicted the opposite trend, with elongation occurring at faster rather than slower speeds. This also has some empirical support, such as the work done by Pitcher & Wyche [14] that found school shape to be more spherical when fish were motionless compared to ellipsoid when they began stable cruising behaviour. Breder [13] observed that the elongated body shape of fish results in individuals reducing distances to neighbours located on either side compared to neighbours in front or behind, a phenomenon that has been observed by other empirical work on fish [2,15]. Breder [13] hypothesized that this was the result of swimming movement occurring along the horizontal plane, necessitating greater distances along the axis of motion. This would ultimately lead to more elongated schools at faster speeds as individuals require greater reaction distances.

Consequently, our understanding of the interactions between individuals and their effect on global group structure is to some extent contradictory. This hinders our ability to understand how information spreads between individuals, which is critical to our understanding of the collective functioning of groups. For instance, Attanasi *et al.* [16] found faster information transfer within highly polarized groups, meaning that individuals derive an important functional benefit when they closely align with their neighbours as it allows the group to respond quickly and cohesively to real-time changes in their environment. Despite these interesting results, only a few studies have applied information theoretic measures such as transfer entropy to animal collectives or models thereof [17–20]. While these studies yielded vital insight into the process of information transfer across groups, underscoring the need to continue applying information theoretic measures to animal collectives, the current study aimed to elucidate the specific implications of speed, polarization and positional composition on the benefits of grouping, which has not been specifically studied.

In the present experiment, we sought to test not only how speed interacts with polarization, but how this tight relationship impacts positional composition to produce the elongated fish schools observed throughout nature. By analysing five closely related species of fish, we also determined whether these trends were qualitatively consistent across species. And finally, using a novel information theoretic approach, we characterized information flow between loosely polarized shoals travelling at slow speeds to highly polarized schools travelling at fast speeds. This research is crucial as it provides

much needed empirical insight into the trends producing group-wide movement patterns and the potential fitness implications with regard to information flow.

# 2. Material and methods

## 2.1. Study animals

For this experiment, we used five species of rainbowfish from the family Melanotaenia (electronic supplementary material, table S1). These species are all freshwater fish endemic to Australia and the Indo pacific. We used *M. mccullochii* from Skull Creek in Queensland (hereafter SC), *M. nigrans* from George Creek in the Northern Territory (hereafter GC), aquarium bred *M. duboulayi* (hereafter MD), *M. sp* from Burton's Creek in the Northern Territory (hereafter BR) and *M. sp* from Bindoola Creek in Western Australia (hereafter BN). All fish were sourced from wild populations and kept for a minimum of two generations in an aquarium facility in the Northern Territory. Fish were transported to the aquarium facility at the University of Sydney where they were placed in 180 L stock tanks maintained at 23°C with a 12 L : 12 D cycle and fed fish flake ad libitum each evening. All fish were given a minimum of two weeks to acclimate to laboratory conditions before being used in experiments.

## 2.2. Experimental protocol

Trials were conducted in a white Perspex tank filled to a depth of 70 mm. We created an annulus within this tank, which promotes continuous swimming within trials. This was made using two circular white plastic inserts. The outer circle had a diameter of 1480 mm and the inner circle had a diameter of 638 mm. The tank was surrounded on all sides by a white poster board to reduce the influence of external stimuli. There was no covering on the top of the experimental arena, and the arena was lit with fluorescent ceiling lights.

All trials were conducted between the hours of 8.00 and 12.00 to limit any impact that time of day may have and to standardize nutritional state as much as possible [21]. For each replicate, groups of five fish were netted carefully out of their stock tank and placed in a small 1 L holding tank. Fish were then transported to the arena and poured gently into the annulus. To allow fish to fully acclimate to the tank, we gave them a full hour in the tank before filming groups for 5 min. Video was acquired remotely using a Panasonic HC × 1000 suspended 2 m above the test tank filming at 50 fps at a resolution of 1080 dpi.

After each trial, fish were removed from the annulus, photographed and placed in a used stock tank to ensure no individual was used twice. These photos were then opened within ImageJ and used to calculate standard length (SL) for each fish. The average SL for BN was $5.92 \pm 0.83$ cm (mean $\pm$ s.d.), $3.87 \pm 0.37$ cm for BR, $5.31 \pm 0.59$ cm for SC, $5.95 \pm 0.63$ cm for GC and $4.79 \pm 0.54$ cm for MD.

## 2.3. Data extraction

Videos were filmed in HD at 50 fps and analysed using the idTracker package [22]. This tracking software generated $X$ and $Y$ coordinates for each fish through all 15 000 frames. $X$ and $Y$ coordinates were then converted to mm using a ratio of known distance in mm divided by pixels. Given the high frame rate, a moving average smoothing function spanning 10 frames (0.2 s) was used to remove spurious fluctuations in position. These smoothed coordinates were used to calculate speed, measured in body lengths per second using the population average, which was done to account for size differences between species. These measures of speed formed the basis of our analysis of speed-mediated properties of schooling behaviour.

To understand how group morphology and polarization relate to speed, we calculated alignment (average deviation in the angle between each fish and its nearest neighbour, NN) and proportion of the group in front or behind each individual. These calculations were generated based on individual speed, ranging from 0.25 to 5.75 BL s$^{-1}$ in 0.5 BL s$^{-1}$ intervals. Fish were considered to be in front or behind a focal individual when their $x, y$ coordinates fell within either of the 90° zones extending to the front and to the back of the focal individual (dividing lines created at 45°, 135°, 225° and 315° relative to the centre of mass of the focal individual). Polarization and proportion in front or behind was averaged across each trial and then combined to produce a species average.

To investigate how speed and alignment affect information transfer within groups, we first created three speed and alignment categories. These categories were created by dividing the distribution of speed and alignment within each species into thirds. The speed cut off points between tertiles for each species were 3.45 and 4.80 BL s$^{-1}$ for BN, 0.93 and 2.35 BL s$^{-1}$ for BR, 1.49 and 3.27 BL s$^{-1}$ for MD, 1.10 and 3.82 BL s$^{-1}$ for GC and 1.05 and 2.16 BL s$^{-1}$ for SC. The alignment cut off points between tertiles for each species were 0.76 and 0.977 for BN, 0.54 and 0.88 for BR, 0.61 and 0.91 for MD, 0.64 and 0.96 for GC and 0.62 and 0.93 for SC. Within each species' speed and alignment tertiles, we selected the three longest continuous trajectory segments, discarding any in which a given speed or alignment was not maintained for longer than 1 s. Given that there were no trajectory segments in which fish maintained medium speeds for longer than 1 s, there were only two speed categories in the final analysis (slow and fast). For consistency, we also redacted the medium segment of alignment from the analysis. Ultimately, this left us with a total of 120 trajectory segments across all species within the lowest speed tertile, 146 trajectory segments within the highest speed tertile, 150 trajectory segments within the lowest alignment tertile and 145 trajectory segments within the highest alignment tertile.

## 2.4. Calculating transfer entropy

Each of these trajectory segments was used to calculate transfer entropy using methods described in [20]. Transfer entropy is a way of measuring information flow longitudinally between pairs of time-series processes by quantifying how knowledge of one individual's time series reduces the uncertainty in predicting another individual's time series [23,24]. Following [20], we calculate transfer entropy here as the average conditional mutual information about the target's current heading update $x_n$ at time $n$ gained from the heading $y_{n-1}$ of the source at time $n-1$ (relative to the target) given a vector of the $k$ previous relative headings of the target $x_{n-1}^{(k)} = \{x_{n-1}, x_{n-2} \ldots x_{n-k}\}$:

$$T_{Y \to X} = \left\langle \log \frac{p(x_n | x_{n-1}^k, y_n)}{p(x_n | x_{n-1}^k)} \right\rangle.$$

For all trajectories within each speed and alignment category, transfer entropy was calculated as an average over samples from all relevant directed pairs of individuals within the group. Transfer entropy was calculated using the KSG estimator [25] with the JIDT software [26], using four nearest samples in the search space. We used a target embedding history length of $k = 3$, which was set to minimize information from target past attributed as a transfer, and a source-target delay of 100 ms (five time steps). This process produces one measure for each trajectory segment of mean group pairwise transfer entropy across all directed pairs within the group. To test whether there was a statistically significant directed relationship between source and target, we compared our estimates of transfer entropy to surrogate distributions, which were calculated by randomizing the order of relative headings within each source trajectory segment and computing average transfer entropy for the resulting sample (techniques described in Lizier [26]). Finally, note that while local or pointwise transfer entropy for each given sample for a directed pair at one specific time may be positive or negative (see [20] for details), the average transfer entropies $T_{Y \to X}$ for each trajectory segment over these local values should, in theory, be non-negative. In practice, the bias correction feature of the KSG estimator can give rise to small negative average values for some trajectory segments; these should be interpreted as being consistent with the surrogate distribution.

## 2.5. Statistical analysis

Using the lme package in R [27], we created mixed effect models to investigate the effect of speed on polarization and positioning. Each response variable (polarization and positioning) was separately tested against the orthogonal first- and second-order polynomials of speed (R code available in SI). This was done to investigate whether the quadratic term significantly improved the regression compared to the linear term. To analyse general trends across species, we included species as a random effect. We then used 95% confidence intervals, marginal and conditional $R^2$ values to investigate the significance of both the main effect, speed, and the random effect, species. Mixed effect models were also used to compare transfer entropy (nats) between low speeds and high speeds, as well as between low alignment and high alignment.

To visualize the relationship between speed and relative neighbour positioning, density heatplots were created for each speed tertile. This was done by anchoring each fish to the origin of the heat plot in turn and recording the relative position of neighbours when group centroids were moving slow, medium or fast, using the previously defined tertiles. For each heat plot, warmer colours denote higher encounter frequencies.

# 3. Results

Heatplots revealed a speed-mediated shift in neighbour positioning that was qualitatively consistent across all five species. Generally, there was a tendency for neighbour position to transition from the sides to the front and back as speeds increased (figure 1).

This transition from side-by-side group positioning to linear in front-behind positioning has a quadratic relationship with the proportion of individuals in front and behind the focal individual increasing from 0.5 at slow speeds, indicating no preference for specific spatial positioning, to greater than 0.65 at faster speeds, indicating a strong preference for neighbours in front and behind. Our mixed effect model revealed a significant effect of speed$^2$ (mixed effect model: $F = 161.17$, $p < 0.001$; 95% confidence interval: $-0.17$ to $-0.07$, figure 2). The random effect had a confidence interval of 0.03 to 0.1 and improved the fit of the model from a marginal $R^2$ value of 0.48 to a conditional $R^2$ value of 0.92 (for species-specific trendlines, see electronic supplementary material, figure S1).

Along with this speed-mediated shift in group positioning, there was also a speed-mediated increase in alignment with NN. As speeds increased, the angular deviation with NN decreased, though this plateaued at faster speeds, likely because there is a hard limit on how polarized groups can be, producing a saturating effect. Our mixed effect models revealed a significant effect of speed$^2$ (mixed effect model: $F = 553.16$, $p < 0.001$; 95% confidence interval: 47.16 to 64.31; figure 3). The random effect had a confidence interval of 2.80 to 10.32 and slightly improved the fit of the model from a marginal $R^2$ value of 0.87 to a conditional $R^2$ value of 0.95.

Information transfer existed between groups of fish when moving at both slow and fast speeds and low and high alignment, as determined by the transfer entropy being greater than the surrogate distribution + 2*s.d. Transfer entropy varied significantly between slow-moving and fast-moving fish (mixed effect model: $F = 22.12$, $p < 0.001$; 95% confidence interval: 0.007 to 0.02; figure 4), with greater information flow occurring at fast speeds compared to slow speeds. The random effect, species, had a confidence interval of 0.001 to 0.01 and slightly improved the fit of the model from a marginal $R^2$ value of 0.19 to a conditional $R^2$ value of 0.25. Transfer entropy also varied significantly based on whether fish had a low alignment or high alignment (mixed effect model: $F = 56.66$, $p < 0.001$; 95% confidence interval: 0.014 to 0.24; figure 5). Species had a confidence interval of 0.002 to 0.01 and slightly improved the fit of the model from a marginal $R^2$ value of 0.35 to a conditional $R^2$ value of 0.41.

# 4. Discussion

Our results underscore the importance of speed in mediating both local interactions and global patterns. At the local level, increases in speed led to increases in alignment and shifts from lateral to linear neighbour positioning. At the global level, these increases in linear neighbour positioning resulted in elongation of the group. Furthermore, mean pairwise transfer entropy increased with speed and alignment, implying an adaptive value to forming faster, more polarized and linear groups. Ultimately, this research provides vital empirical insight into the mechanisms underlying elongation of moving animal groups and highlights the functional significance of cohesive and coordinated movement.

Contrary to the theoretical work done by Hemelrijk & Hildenbrandt [28], we found increasing elongation at increasing speeds. This seems to be the result of both elongated zones of interaction and a shift from favouring lateral neighbour positioning to linear neighbour positioning. In accordance with Breder [13] and Katz et al. [2], we found smaller distances between lateral neighbours compared to neighbours located in front or behind the focal individual. This elongation became more pronounced with speed, likely due to the need for greater reaction distances along the axis of motion. However, an important and somewhat surprising outcome of this study is the shift we found from lateral to front–back neighbour positioning at increasing speeds. We found that this shift in positional preferences had a quadratic relationship with speed. Individuals increasingly favoured leader–follower formations as speeds increased, though this preference plateaued at greater speeds, potentially signalling a stabilization in spatial positioning at greater speeds. However, recent work by Ashraf et al. [29] found that red-nosed

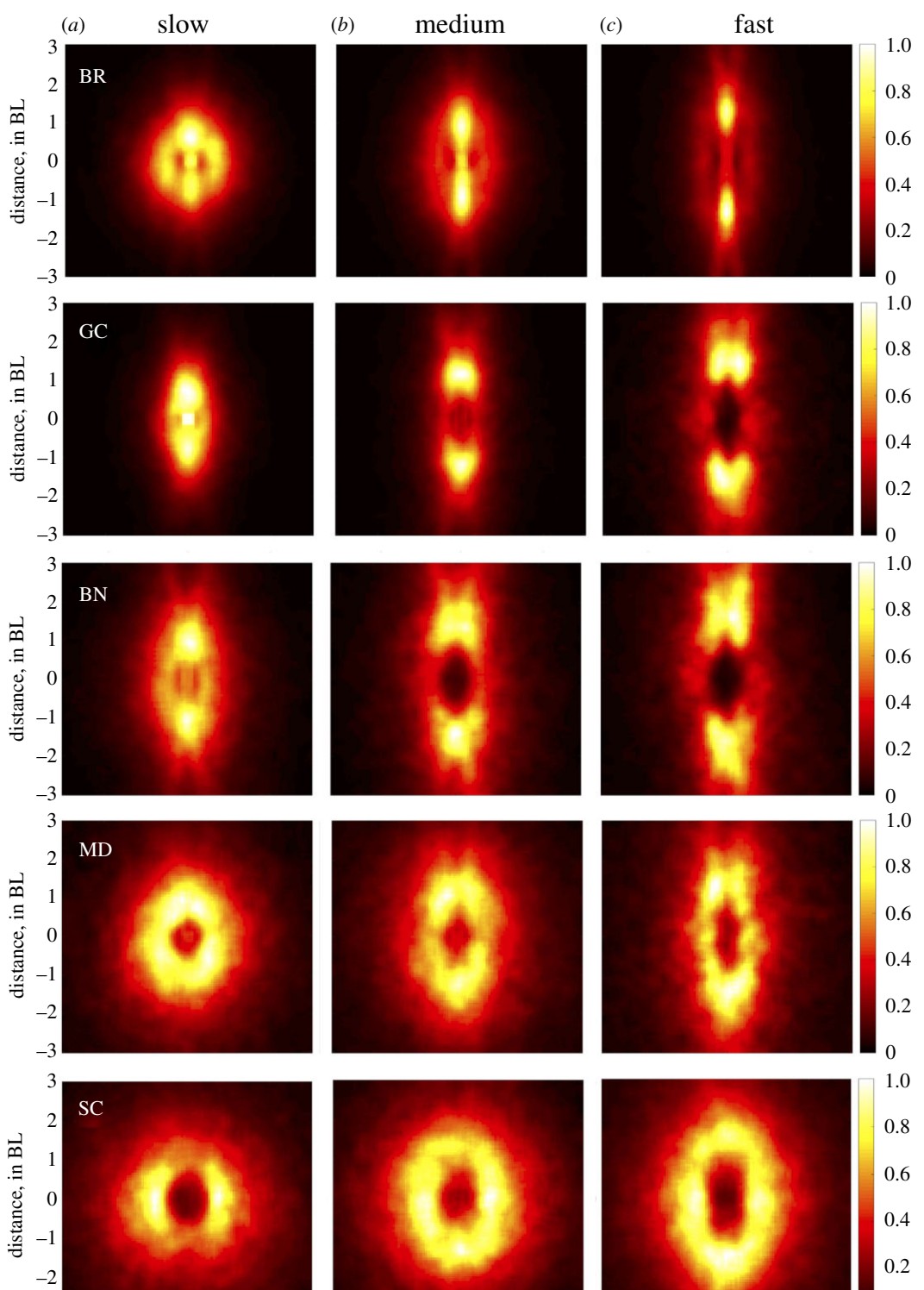

**Figure 1.** Heatplots of positional frequency when groups were moving at slow (*a*), medium (*b*) and fast (*c*) speeds for each of our five species (top to bottom: BR, GC, BN, MD, SC). Speed categories were derived from the speed distributions within each species and normalized by the maximal density value. Focal individuals are at the origin facing positive along the *y*-axis, with warmer colours denoting the higher frequency of encounters.

tetras forced to swim faster than their preferred free-swimming velocities shifted from diamond formations to phalanx formations. It is possible that the current study would have found a subsequent decrease in linear neighbour positioning had the experimental set-up not relied solely on elective group swimming speeds.

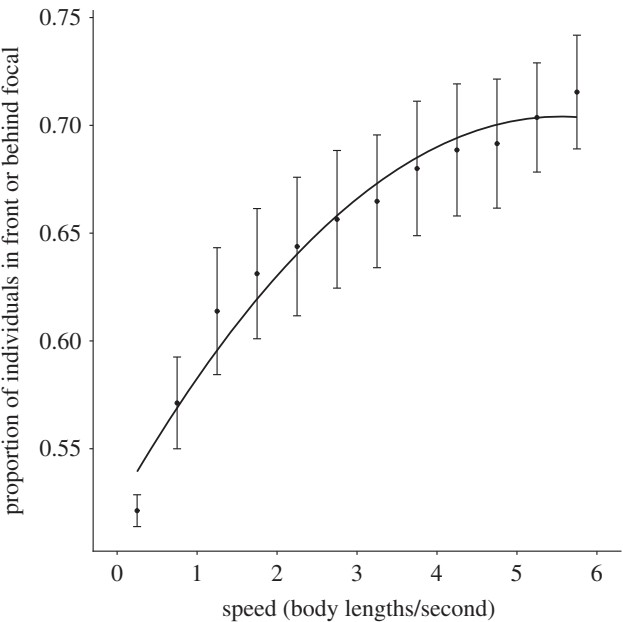

**Figure 2.** Graph showing the average proportion of individuals in front or behind a focal individual $\pm$ s.e. bars as a function of speed (BL s$^{-1}$) with a quadratic trendline. As speed increases, the proportion of individuals positioned in front or behind the focal individual increases sharply initially before beginning to level off.

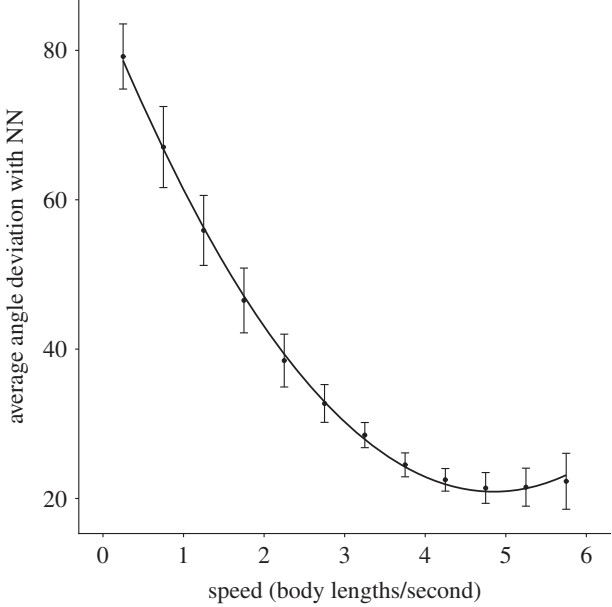

**Figure 3.** Graph showing average angle deviation with NN $\pm$ s.e. bars as a function of speed (BL s$^{-1}$) with a quadratic trendline. As speed increases, deviation in angle with NN decreases, meaning that overall group alignment increases.

Within the observed range of free-swimming speeds, our results point to an increasingly elongated shape as groups transition from low order to high order. Importantly, this shift in neighbour positional preferences was qualitatively similar across all five species.

   This consistent shift towards more linear neighbour positioning across species may imply a common benefit to forming faster, more polarized and elongated groups. While the energetic benefits of swimming in a group are well established (e.g. reduced oxygen consumption [30] and lower tail beat frequency [31] in collectively moving fish compared to solitary fish), there may be further benefits to individuals within specific group formations. Theoretical work by Hemelrijk *et al.* [32] demonstrated how individuals swimming side-by-side in a phalanx formation gained little to no energetic payoff compared to individuals swimming in a linear or diamond formation (however, see [29]). This may

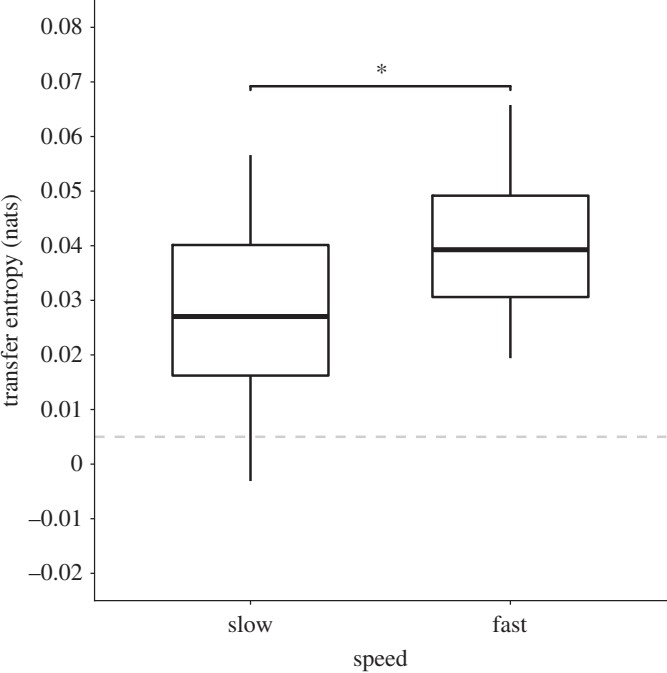

**Figure 4.** Box plot of transfer entropy (nats) when fish were travelling at slow speeds (i.e. those speeds within the lower tertile) or at fast speeds (upper tertile), measured as body lengths per second. Medians and interquartile ranges are shown. The dashed horizontal line shows the null TE + 2*s.d., indicating the threshold above which TE is significantly more than would be expected by chance.

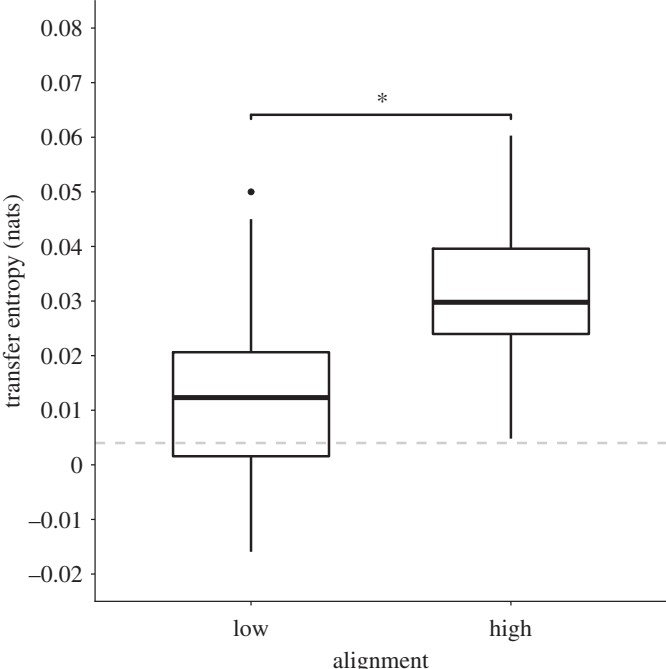

**Figure 5.** Box plot of transfer entropy (nats) when fish had low alignment (lowest tertile) or high alignment (upper tertile), measured as the degree of deviation with NN. Medians and interquartile ranges are shown. The dashed horizontal line shows the null TE + 2*s.d., indicating the threshold above which TE is significantly more than would be expected by chance.

explain why we found a shift in favour of more linear formations at faster speeds as individuals would have offset the increased energy requirements of fast swimming [33] with greater hydrodynamic benefits of more linear formations.

In addition to the putative energetic benefits of this positional shift with speed, we found significantly greater information transfer at faster speeds and higher alignments, indicating a fitness benefit to high-

order groups as they increase the ability to coordinate or synchronize behaviour. Interestingly, low alignment seemed more detrimental to information flow than slow speeds given that transfer entropy between unaligned individuals was only slightly statistically larger than the null distribution. Both speed and alignment, averaged at the group level, are important, which is not surprising as speed and alignment are positively and tightly correlated. These results underscore how individuals might benefit from transitioning away from low alignment and slow speeds to faster, more aligned and linear groups.

To understand how groups move collectively and make decisions, it is first necessary to understand the underlying interaction between individuals. This study provides an important link between individual interactions and group-level functioning, specifically showing increases in the flow of information as groups shift from low order to high order and linear movement. While previous theoretical studies have addressed the importance of density in producing oblong schools, this study has shown that at constant densities, the positional composition can shift to leader–follower formations as a function of speed. Additional research can use the present study's findings in conjunction with school density to understand how both speed and group size might interact. Importantly, this research defines the specific relationship between speed, alignment and positional composition of groups, highlighting the importance of speed in mediating both individual behaviour and group morphology more generally.

Ethics. This study and protocol was approved by the University of Sydney Animal Ethics Committee (Permit Number: 2015/807).

Data accessibility. Summary data and code are available at the Dryad Digital Repository: https://doi.org/10.5061/dryad. bd08ft6 [34]. Full raw dataset is available upon request.

Authors' contributions. M.K. and A.W. designed the experiments, M.K. performed the experiments, all authors performed the analysis and edited the manuscript written by M.K.

Competing interests. We have no competing interests.

Funding. This research was funded by a grant from the Australian Research Council (ARC) and by an Australian Government Research Training Program (RTP) Scholarship.

Acknowledgements. The authors would like to thank Emanuele Crosato and Mikhail Prokopenko for granting access to their code quantifying transfer entropy. The authors would also like to thank Mate Nagy and one anonymous reviewer for their constructive comments that improved the quality of this manuscript.

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
