## [Reviewer comments · Royal Society Open Science]

Review History

RSOS-181482.R0 (Original submission)

Review form: Reviewer 1 (Mate Nagy)

Is the manuscript scientifically sound in its present form?

Yes

Are the interpretations and conclusions justified by the results?

Yes

Is the language acceptable?

Yes

Is it clear how to access all supporting data?

Yes

Do you have any ethical concerns with this paper?

No

Have you any concerns about statistical analyses in this paper?

No

Recommendation?

Accept with minor revision (please list in comments)

Comments to the Author(s)

This paper reports experiments on schooling of groups of 5 fish belonging to 5 closely related species and studies the effect of speed, polarization and elongated shape of the shoal in relation to the information transfer between the individuals. The manuscript is interesting and generally well written. The description of the experiment and the analysis are sound and the results fit well into the existing literature, and on the other hand, aiming to fill a gap in an important but yet not so much explored area. It is also a strength of the manuscript that the authors have done the experiments on multiple (although closely related) species rather than a single selected model. I have only minor comments to be addressed.

The abstract is ambitious and maybe a bit too broad when speaking about groups in the whole animal kingdom but the later parts of the manuscript refer almost exclusively to fish. The relationship between speed and collective motion in animal groups is a hot topic and is in the scope of several recent research papers. This further verifies the importance of the submitted manuscript, although I miss references to some of these works as the bibliography contains very few papers from the last few years. I would be happy to see, for example, Pettit et al. 2015 and Jolles et al. 2017 referred. Along the same line, transfer entropy is a novel method but there are also more examples of papers (including some of the authors own works) applying it to collective animal behaviour as compared to what the text suggests at Line 81-83, e.g., Lord et al. 2016, Orange et al. 2015.

The authors write in the Methods – Data extraction section that “we selected the 3 longest continuous trajectory segments”. It is not clear to me what happened after this selection. Was this selection done for each trial of each group separately? How was the sample size derived for the statistical tests? Please specify it explicitly.

I’ve found as supporting data only the derived metrics. Are the full trajectories available either at the same repository or on request? That would be essential if someone would like to reproduce the results or calculate comparable statistics.

To summarize, this is a great manuscript and I think it will be of interests of the readership of Royal Society Open Science ranging from experts studying animal behaviour to researchers outside of our field. As a consequence, I recommend publication after minor revisions.

Mate Nagy

Max Planck Institute for Ornithology, Department of Collective Behaviour &
Department of Biology, Konstanz University, Konstanz, Germany
MTA-ELTE Statistical and Biological Physics Research Group, Budapest, Hungary

References:

J Jolles, NJ Boogert, VH Sridhar, ID Couzin, A Manica (2017) Consistent Individual Differences Drive Collective Behavior and Group Functioning of Schooling Fish, *Current Biology* 27 (18), 2862-2868

WM Lord, J Sun, NT Ouellette, EM Bollt (2016) Inference of Causal Information Flow in

Collective Animal Behavior, in IEEE Transactions on Molecular, Biological and Multi-Scale Communications 2 (1), 107-116.

N. Orange, N. Abaid (2015) A transfer entropy analysis of leader-follower interactions in flying bats, Eur. Phys. J. Spec. Top. 224 (17-18), 3279-3293

B. Pettit, Z. Akos, T. Vicsek, D. Bior (2015) Speed Determines Leadership and Leadership Determines Learning during Pigeon Flocking, Current Biology 25 (23), 3134-4147

Review form: Reviewer 2

Is the manuscript scientifically sound in its present form?

No

Are the interpretations and conclusions justified by the results?

Yes

Is the language acceptable?

Yes

Is it clear how to access all supporting data?

Yes

Do you have any ethical concerns with this paper?

No

Have you any concerns about statistical analyses in this paper?

I do not feel qualified to assess the statistics

Recommendation?

Major revision is needed (please make suggestions in comments)

Comments to the Author(s)

Review of the paper RSOS-181482:

Speed-mediated properties of schooling

By Kent, Lukeman, Lizier, Ward

Summary:

This paper reports on an experimental study on the collective dynamics of swimming fish, using 5 species of rainbowfish from the Melanoteaenia family. The experiments consist of free swimming trials of groups of 5 fish in an annular swimming arena. The authors use video recordings to discuss the fish school morphology and the information transfer within the group as a function of the swimming speed.

Overall I find the paper poses an interesting question and brings valuable experimental results on a subject where quantitative experimental data is scarce. I have however a few comments that should be addressed before publication.

Comments:

- Much more details on the experimental setup/protocol/results are needed. I can suggest: A figure of the experimental setup would be helpful; examples of typical tracking for each species and why not supplementary videos showing a typical experiment for each species; a table summarizing species name, abbreviation, standard length, speed and alignment cut-offs,...
- About transfer entropy: The specific definition used in the paper should be included (Is it Eq. 2 from the Crosato et al. 2017 paper?). Comment on the values of the local transfer entropies before averaging (the issue of negative local transfer entropy and misinformation).
- Statistical analyses: Please include a brief explanation of the Mixed-Effects Model. "using the lme package in R" is not sufficiently clear.
- About the quadratic fits (it should be written in the captions of figures 2 and 3 that the line is a quadratic fit): Do the authors have any interpretation about the "quadratic" nature of the curves in Figs 2 and 3? If nothing is said about the reason for things being quadratic we might as well keep "increasing and decreasing trends" for Figs. 2 and 3, respectively. Actually in Figure 2 two linear curves would fit better, with a slope change at 1 BL/s.
- Recent works by Ashraf et al. (refs below) with *Hemigrammus bleheri* have shown a tendency of fish to align side-by-side and to reduce the nearest-neighbor-distance when fish are swimming faster. That observation is different from what is reported in the present paper. Although the experimental setups are different (Ashraf et al. impose a swimming speed using an external flow), a comment on those papers should be included.

Refs:

Ashraf, I., et al. "Synchronization and collective swimming patterns in fish (*Hemigrammus bleheri*).*" Journal of The Royal Society Interface* 13.123 (2016): 20160734.

Ashraf, I., et al. "Simple phalanx pattern leads to energy saving in cohesive fish schooling." *Proceedings of the National Academy of Sciences* 114.36 (2017): 9599-9604.

- In lines 208-210 the authors write: "As speeds increased, angular deviation with NN decreased, though this plateaued at faster speeds, likely due to the fact that there is a hard limit on how polarised groups can be, producing a saturating effect." Please explain, what is the explicit hard limit?

Decision letter (RSOS-181482.R0)

19-Nov-2018

Dear Dr Kent,

The editors assigned to your paper ("Speed-mediated properties of schooling") have now received comments from reviewers. We would like you to revise your paper in accordance with the referee and Associate Editor suggestions which can be found below (not including confidential reports to the Editor). Please note this decision does not guarantee eventual acceptance.

Please submit a copy of your revised paper before 12-Dec-2018. Please note that the revision deadline will expire at 00.00am on this date. If we do not hear from you within this time then it will be assumed that the paper has been withdrawn. In exceptional circumstances, extensions may be possible if agreed with the Editorial Office in advance. We do not allow multiple rounds of revision so we urge you to make every effort to fully address all of the comments at this stage. If deemed necessary by the Editors, your manuscript will be sent back to one or more of the original reviewers for assessment. If the original reviewers are not available, we may invite new reviewers.

- Data accessibility

<http://datadryad.org/submit?journalID=RSOS&manu=RSOS-181482>

- Competing interests

- Authors' contributions

All submissions, other than those with a single author, must include an Authors' Contributions section which individually lists the specific contribution of each author. The list of Authors

should meet all of the following criteria; 1) substantial contributions to conception and design, or acquisition of data, or analysis and interpretation of data; 2) drafting the article or revising it critically for important intellectual content; and 3) final approval of the version to be published.

- Acknowledgements

- Funding statement

Please note that Royal Society Open Science charge article processing charges for all new submissions that are accepted for publication. Charges will also apply to papers transferred to Royal Society Open Science from other Royal Society Publishing journals, as well as papers submitted as part of our collaboration with the Royal Society of Chemistry (<http://rsos.royalsocietypublishing.org/chemistry>). If your manuscript is newly submitted and subsequently accepted for publication, you will be asked to pay the article processing charge, unless you request a waiver and this is approved by Royal Society Publishing. You can find out more about the charges at <http://rsos.royalsocietypublishing.org/page/charges>. Should you have any queries, please contact openscience@royalsociety.org.

Kind regards,

Royal Society Open Science Editorial Office
Royal Society Open Science
openscience@royalsociety.org

on behalf of Professor Brooke Flammang (Associate Editor) and Professor Kevin Padian (Subject Editor)
openscience@royalsociety.org

Associate Editor's comments (Professor Brooke Flammang):

Dear Authors,

Both reviewers find the paper to be interesting and scientifically strong, yet lacking in a number of important details. Please see both reviewers comments for specific suggestions to improve your manuscript.

Comments to Author:

Reviewers' Comments to Author:

Reviewer: 1

Comments to the Author(s)

This paper reports experiments on schooling of groups of 5 fish belonging to 5 closely related species and studies the effect of speed, polarization and elongated shape of the shoal in relation to the information transfer between the individuals. The manuscript is interesting and generally well written. The description of the experiment and the analysis are sound and the results fit well into the existing literature, and on the other hand, aiming to fill a gap in an important but yet not so much explored area. It is also a strength of the manuscript that the authors have done the experiments on multiple (although closely related) species rather than a single selected model. I have only minor comments to be addressed.

The abstract is ambitious and maybe a bit too broad when speaking about groups in the whole animal kingdom but the later parts of the manuscript refer almost exclusively to fish. The relationship between speed and collective motion in animal groups is a hot topic and is in the scope of several recent research papers. This further verifies the importance of the submitted manuscript, although I miss references to some of these works as the bibliography contains very few papers from the last few years. I would be happy to see, for example, Pettit et al. 2015 and Jolles et al. 2017 referred. Along the same line, transfer entropy is a novel method but there are also more examples of papers (including some of the authors own works) applying it to collective animal behaviour as compared to what the text suggests at Line 81-83, e.g., Lord et al. 2016, Orange et al. 2015.

The authors write in the Methods – Data extraction section that “we selected the 3 longest continuous trajectory segments”. It is not clear to me what happened after this selection. Was this selection done for each trial of each group separately? How was the sample size derived for the statistical tests? Please specify it explicitly.

I've found as supporting data only the derived metrics. Are the full trajectories available either at the same repository or on request? That would be essential if someone would like to reproduce the results or calculate comparable statistics.

To summarize, this is a great manuscript and I think it will be of interests of the readership of Royal Society Open Science ranging from experts studying animal behaviour to researchers outside of our field. As a consequence, I recommend publication after minor revisions.

Mate Nagy

Max Planck Institute for Ornithology, Department of Collective Behaviour &
Department of Biology, Konstanz University, Konstanz, Germany
MTA-ELTE Statistical and Biological Physics Research Group, Budapest, Hungary

References:

J Jolles, NJ Boogert, VH Sridhar, ID Couzin, A Manica (2017) Consistent Individual Differences Drive Collective Behavior and Group Functioning of Schooling Fish, *Current Biology* 27 (18), 2862-2868

WM Lord, J Sun, NT Ouellette, EM Bollt (2016) Inference of Causal Information Flow in

Collective Animal Behavior, in IEEE Transactions on Molecular, Biological and Multi-Scale Communications 2 (1), 107-116.

N. Orange, N. Abaid (2015) A transfer entropy analysis of leader-follower interactions in flying bats, Eur. Phys. J. Spec. Top. 224 (17-18), 3279-3293

B. Pettit, Z. Akos, T. Vicsek, D. Bior (2015) Speed Determines Leadership and Leadership Determines Learning during Pigeon Flocking, Current Biology 25 (23), 3134-4147

Reviewer: 2

Comments to the Author(s)

Review of the paper RSOS-181482:

Speed-mediated properties of schooling

By Kent, Lukeman, Lizier, Ward

Summary:

This paper reports on an experimental study on the collective dynamics of swimming fish, using 5 species of rainbowfish from the Melanoteaenia family. The experiments consist of free swimming trials of groups of 5 fish in an annular swimming arena. The authors use video recordings to discuss the fish school morphology and the information transfer within the group as a function of the swimming speed.

Overall I find the paper poses an interesting question and brings valuable experimental results on a subject where quantitative experimental data is scarce. I have however a few comments that should be addressed before publication.

Comments:

- Much more details on the experimental setup/protocol/results are needed. I can suggest: A figure of the experimental setup would be helpful; examples of typical tracking for each species and why not supplementary videos showing a typical experiment for each species; a table summarizing species name, abbreviation, standard length, speed and alignment cut-offs,...

- About transfer entropy: The specific definition used in the paper should be included (Is it Eq. 2 from the Crosato et al. 2017 paper?). Comment on the values of the local transfer entropies before averaging (the issue of negative local transfer entropy and misinformation).

- Statistical analyses: Please include a brief explanation of the Mixed-Effects Model. "using the lme package in R" is not sufficiently clear.

- About the quadratic fits (it should be written in the captions of figures 2 and 3 that the line is a quadratic fit): Do the authors have any interpretation about the "quadratic" nature of the curves in Figs 2 and 3? If nothing is said about the reason for things being quadratic we might as well keep "increasing and decreasing trends" for Figs. 2 and 3, respectively. Actually in Figure 2 two linear curves would fit better, with a slope change at 1 BL/s.

- Recent works by Ashraf et al. (refs below) with *Hemigrammus bleheri* have shown a tendency of fish to align side-by-side and to reduce the nearest-neighbor-distance when fish are swimming

faster. That observation is different from what is reported in the present paper. Although the experimental setups are different (Ashraf et al. impose a swimming speed using an external flow), a comment on those papers should be included.

Refs:

Ashraf, I., et al. "Synchronization and collective swimming patterns in fish (*Hemigrammus bleheri*)." *Journal of The Royal Society Interface* 13.123 (2016): 20160734.

Ashraf, I., et al. "Simple phalanx pattern leads to energy saving in cohesive fish schooling." *Proceedings of the National Academy of Sciences* 114.36 (2017): 9599-9604.

- In lines 208-210 the authors write: "As speeds increased, angular deviation with NN decreased, though this plateaued at faster speeds, likely due to the fact that there is a hard limit on how polarised groups can be, producing a saturating effect." Please explain, what is the explicit hard limit?

Author's Response to Decision Letter for (RSOS-181482.R0)

See Appendix A.

Decision letter (RSOS-181482.R1)

23-Jan-2019

Dear Dr Kent,

I am pleased to inform you that your manuscript entitled "Speed-mediated properties of schooling" is now accepted for publication in Royal Society Open Science.

on behalf of Professor Brooke Flammang (Associate Editor) and Professor Kevin Padian (Subject Editor)
openscience@royalsociety.org

Follow Royal Society Publishing on Twitter: [@RSocPublishing](https://twitter.com/RSocPublishing)

Appendix A

Associate Editor's comments (Professor Brooke Flammang):

Dear Authors,

Both reviewers find the paper to be interesting and scientifically strong, yet lacking in a number of important details. Please see both reviewers comments for specific suggestions to improve your manuscript.

Reviewer: 1

Comments to the Author(s)

This paper reports experiments on schooling of groups of 5 fish belonging to 5 closely related species and studies the effect of speed, polarization and elongated shape of the shoal in relation to the information transfer between the individuals. The manuscript is interesting and generally well written. The description of the experiment and the analysis are sound and the results fit well into the existing literature, and on the other hand, aiming to fill a gap in an important but yet not so much explored area. It is also a strength of the manuscript that the authors have done the experiments on multiple (although closely related) species rather than a single selected model. I have only minor comments to be addressed.

The abstract is ambitious and maybe a bit too broad when speaking about groups in the whole animal kingdom but the later parts of the manuscript refer almost exclusively to fish. The relationship between speed and collective motion in animal groups is a hot topic and is in the scope of several recent research papers. This further verifies the importance of the submitted manuscript, although I miss references to some of these works as the bibliography contains very few papers from the last few years. I would be happy to see, for example, Pettit et al. 2015 and Jolles et al. 2017 referred. Along the same line, transfer entropy is a novel method but there are also more examples of papers (including some of the authors own works) applying it to collective animal behaviour as compared to what the text suggests at Line 81-83, e.g., Lord et al. 2016, Orange et al. 2015.

The authors write in the Methods – Data extraction section that “we selected the 3 longest continuous trajectory segments”. It is not clear to me what happened after this selection. Was this selection done for each trial of each group separately? How was the sample size derived for the statistical tests? Please specify it explicitly.

I've found as supporting data only the derived metrics. Are the full trajectories available either at the same repository or on request? That would be essential if someone would like to reproduce the results or calculate comparable statistics.

To summarize, this is a great manuscript and I think it will be of interests of the readership of Royal Society Open Science ranging from experts studying animal behaviour to researchers outside of our field. As a consequence, I recommend publication after minor revisions.

Mate Nagy

Max Planck Institute for Ornithology, Department of Collective Behaviour & Department of Biology, Konstanz University, Konstanz, Germany MTA-ELTE Statistical and Biological Physics Research Group, Budapest, Hungary

Responses to Reviewer 1

The referee makes a good point that the abstract could be pared down to better reflect the rest of the paper. Accordingly, we have rewritten lines 19-20 to reflect our use of fish in this experiment.

As Reviewer 1 points out, the relationship between speed and collective motion is currently a hot topic and we agree that our manuscript would benefit from the inclusion of recent work by Pettit et al. and Jolles et al. that discuss the importance of speed in mediating leadership and various group dynamics. We have added these references to the manuscript on lines 41-42. We believe, as the referee suggested, that the inclusion of these papers serves to underscore the importance of the current manuscript and the research presented. As also suggested, we added references to other works that measured TE in animal models and rewrote lines 83-88 to highlight the specific application of information theory in the current study and the ways in which our application differs slightly from previous work.

To clarify the methods, we have reworded lines 157-158 and provided more explicit information pertaining to the sample sizes on lines 162-164.

While we have not made the raw trajectories available through the online repository site Dryad, we have added a sentence to the data availability section specifying that the full trajectory files will be available upon request. Each trial is a separate .mat or .txt file consisting of 15 columns and 15,000 rows.

Reviewer: 2

Comments to the Author(s)

Summary: This paper reports on an experimental study on the collective dynamics of swimming fish, using 5 species of rainbowfish from the Melanoteaenia family. The experiments consist of free swimming trials of groups of 5 fish in an annular swimming arena. The authors use video recordings to discuss the fish school morphology and the information transfer within the group as a function of the swimming speed.

Overall I find the paper poses an interesting question and brings valuable experimental results on a subject where quantitative experimental data is scarce. I have however a few comments that should be addressed before publication.

Comments:

- Much more details on the experimental setup/protocol/results are needed. I can suggest: A figure of the experimental setup would be helpful; examples of typical tracking for each species and why not supplementary videos showing a typical experiment for each species; a table summarizing species name, abbreviation, standard length, speed and alignment cut-offs,...

- About transfer entropy: The specific definition used in the paper should be included (Is it Eq. 2 from the Crosato et al. 2017 paper?). Comment on the values of the local transfer entropies before averaging (the issue of negative local transfer entropy and misinformation).

- Statistical analyses: Please include a brief explanation of the Mixed-Effects Model. "using the lme package in R" is not sufficiently clear.

- About the quadratic fits (it should be written in the captions of figures 2 and 3 that the line is a quadratic fit): Do the authors have any interpretation about the "quadratic" nature of the curves in Figs 2 and 3? If nothing is said about the reason for things being quadratic we might as well keep "increasing and decreasing trends" for Figs. 2 and 3, respectively. Actually in Figure 2 two linear curves would fit better, with a slope change at 1 BL/s.

- Recent works by Ashraf et al. (refs below) with *Hemigrammus bleheri* have shown a tendency of fish to align side-by-side and to reduce the nearest-neighbor-distance when fish are swimming faster. That observation is different from what is reported in the present paper. Although the experimental setups are different (Ashraf et al. impose a swimming speed using an external flow), a comment on those papers should be included.

- In lines 208-210 the authors write: "As speeds increased, angular deviation with NN decreased, though this plateaued at faster speeds, likely due to the fact that there is a hard limit on how polarised groups can be, producing a saturating effect." Please explain, what is the explicit hard limit?

Responses to Reviewer 2

As Reviewer 2 suggested, we have made short excerpts of tracked video from each species available through the online repository site Dryad. This will be helpful in demonstrating our methods to any interested readers. We have also included a table of species names, abbreviations, SLs and speed/alignment cut-offs in the SI, with an in-text reference to Table S1 on line 100.

We appreciate reviewer 2's enquiries regarding transfer entropy and agree that our methods could be more specific. As suggested, we have significantly expanded our presentation of how we calculated transfer entropy and have provided the specific equation rather than simply referring the reader to Crosato et al. (see the new transfer entropy section in the methods on lines 166-191). Furthermore, at the end of this expanded section we have included a brief comment on "local transfer entropies" being positive or negative as the reviewer suggests. We have deferred the reader to Crosato for full details on this particular point because the local transfer entropy values within each segment are not dealt with in this paper (only the averages are). We note that the reader may confuse negative local transfer entropies with the negative average transfer entropies for some trajectory segments visible in Figure 4 and 5, and so we have added further clarification at this point in the manuscript on why such negative average values may occur and what they mean.

The referee makes a good point regarding our explanation of mixed effect models. We have therefore added greater explanation on lines 194-196 and provided an in-text reference to our R code, which is available through the online repository site Dryad (line 196).

Reviewer 2 also makes a good point about the need to discuss the specific quadratic relationship reported between linear neighbour positioning and speed. On lines 282-288 in the discussion, we have provided greater interpretation of our quadratic fit as well as discussed this in the context of the work done by Ashraf et al. Regarding the quadratic relationship between alignment and speed, there is a hard limit at 0 degrees because that is the point at which individuals are perfectly aligned. Therefore, the trend naturally plateaus as individuals approach near perfect alignment with neighbours.